# Ineffectiveness of Antiresorptive Agent Drug Holidays in Osteoporosis Patients for Treatment of Medication-Related Osteonecrosis of the Jaw: Consideration from Immunohistological Observation of Osteoclast Suppression and Treatment Outcomes

**DOI:** 10.3390/ijerph191710898

**Published:** 2022-09-01

**Authors:** Keisuke Omori, Mitsunobu Otsuru, Kota Morishita, Saki Hayashida, Koki Suyama, Tomofumi Naruse, Sakiko Soutome, Masahiro Umeda

**Affiliations:** 1Department of Clinical Oral Oncology, Nagasaki University Graduate School of Biomedical Sciences, Nagasaki 852-8588, Japan; 2Department of Oral Health, Nagasaki University Graduate School of Biomedical Sciences, Nagasaki 852-8588, Japan

**Keywords:** antiresorptive agents, bisphosphonate, denosumab, drug holiday, osteoclast suppression

## Abstract

In patients with osteoporosis receiving antiresorptive agents (ARs), it has been widely practiced to withdraw ARs for several months before tooth extraction and during treatment if medication-related osteonecrosis of the jaw (MRONJ) develops. This study examined the effects of drug holidays on recovery from osteoclast suppression and the treatment outcomes. The relationship between the period of the drug holidays and treatment outcomes was examined retrospectively in 166 osteoporosis patients with MRONJ who received ARs. Histological examinations using hematoxylin and eosin staining and cathepsin K stains were performed to observe the recovery from osteoclast suppression in 43 patients in whom living bone was observed in the resection margins of the surgical specimens. Three-month AR drug holidays were not significantly correlated with the treatment outcomes of the 139 patients who underwent surgical treatment and the 27 who underwent conservative treatment. Of the 43 patients who underwent histological investigations, 16 had drug holidays from 7 to 678 days. Osteoclast suppression was observed in almost all patients, except in one without a drug holiday and one with a 261-day drug holiday. These findings suggest that AR drug holidays for approximately 3 months neither recover osteoclast suppression nor affect treatment outcomes.

## 1. Introduction

Currently antiresorptive agents (ARs), such as bisphosphonate (BP) and denosumab (DMB), are widely used to reduce bone resorption due to osteoporosis and skeletal-related events from metastatic bone tumors and multiple myeloma [1,2,3,4,5,6]. However, the presence of medication-related osteonecrosis of the jaw (MRONJ) is one of the severe side effects of ARs [7,8,9,10,11]. The American Association of Oral and Maxillofacial Surgeons (AAOMS) position paper in 2014 [7] recommends a 2-month AR drug holiday before tooth extraction for patients with osteoporosis taking ARs for more than 4 years or with risk factors, such as diabetes or steroid administration; although there is not sufficient evidence to determine whether ARs should be withdrawn. The AAOMS position paper was revised in 2022 [12]; the working group was unable to reach a consensus for drug holidays recommendations, which will be addressed on a case-by-case basis. However, there are several negative reports on AR drug holidays before oral surgery, and the necessity and duration of AR drug holidays before oral surgery are controversial [13].

BP is deposited on the bone matrix and taken up by osteoclasts during the bone resorption process, causing apoptosis of osteoclasts by destroying their cytoskeleton and inhibiting bone resorption [14]. BP deposition in the bone matrix is considered to last from several months to several years. In contrast, DMB selectively inhibits the receptor activator of nuclear factor kappa B ligand (RANKL) expressed on osteoblasts, and suppresses osteoclast differentiation, function, and survival, thereby reducing bone resorption [15]. Considering these pharmacokinetic distinctions, there may be a difference in the drug holiday duration required to recover osteoclast function between BP and DMB, although few reports have examined drug holidays and the clinical significance of BP or DMB.

Morishita et al. [16] recently histologically examined the relationship between drug holidays duration and osteoclast suppression in surgical specimens from patients treated with oral BPs and the reported osteoclast suppression was not recovered after an approximate 6-month drug holiday. However, there are no reports examining the relationship between the DMB drug holiday duration and osteoclast suppression. The purpose of this study is to examine whether the withdrawal of ARs recovers osteoclast suppression and whether there are clinical benefits for withdrawal in osteoporosis patients with MRONJ receiving BP and DMB.

## 2. Materials and Methods

### 2.1. Impact of AR Drug Holidays on Treatment Outcomes

A total of 166 osteoporosis patients with MRONJ were treated at the Oral and Maxillofacial Surgery of Nagasaki University Hospital between January 2012 to December 2020, of which 139 were treated with surgical therapy and 27 with conservative therapy. Some of the cases used in this study overlap with some of the cases used in our group’s previous reports on treatment methods for MRONJ, such as reports on treatment methods for maxillary MRONJ [17], reports on the relationship between drug holiday and sequestrum separation [18], and reports on segmental mandibulectomy [19]. Diagnostic criteria for MRONJ are based on the AAOMS Position Paper [7]. Sex, age, site (upper jaw/lower jaw), stage according to the criteria of AAOMS Position Paper [7], AR type (BP, DMB), administration period (<4 years/≥4 years), corticosteroid, diabetes, serum albumin, serum creatinine, treatment methods, AR drug holiday (<3 months/≥3 months), and treatment outcomes were investigated from the medical records. We have made surgical therapy the first choice for MRONJ regardless of the MRONJ stage, but conservative therapy was selected when surgery was not possible due to the patient’s wishes or poor general condition. Conservative therapy included the use of antimicrobial mouthwash, rinsing of fistulas, periodontal pockets, or gaps between exposed necrotic bone and surrounding soft tissue, and administration of oral antimicrobial agents. Surgical therapy included marginal resection such as partial maxillectomy and marginal mandibulectomy, and segmental mandibulectomy (Figure 1).

We have resected not only the sequestrum but some healthy bone around it. The extent of bone resection was planned based on preoperative CT images but was ultimately determined based on the intraoperative findings such as the presence or absence of bleeding from the bone and changes in bone color. Healing was defined as the disappearance of all symptoms, including bone exposure, and the time to healing was also recorded. Non-healing included improved symptoms, unchanged, and worsened symptoms, and the follow-up period was also recorded. Patients whose symptoms disappeared once and were judged to be “healing” but later relapsed were classified as “non-healing”.

The cumulative cure rate with and without drug suspension was obtained using the Kaplan-Meier method and tested by the log-rank test, divided into surgical cases and preserved cases. Furthermore, each variable examined was analyzed for association with healing rates using a univariate Cox regression analysis, followed by a multiple Cox proportional hazard model including variables with *p*-values smaller than 0.1 in the univariate analysis and the drug holidays longer than 3 months as covariates. The relationship between AR drug holidays and the healing rate was determined using the Kaplan-Meier method, separately for conservative and surgical therapy cases. All statistical analyses were performed using SPSS software (version 26.0; Japan IBM Co., Ltd., Tokyo, Japan), and a 2-tailed *p*-value of <0.05 was considered significant.

### 2.2. Histological Investigation of Recovery from Osteoclast Suppression and Drug Holidays

Histopathology sections were decalcified in 0.5 M ethylenediaminetetraacetic acid solution with gentle agitation at 4 °C, postfixed in a fixing solution, dehydrated, and embedded in paraffin to prepare 4-μm-thick sections. Antigen activation in immunostaining was performed in Dako Target Retrieval Solution (DAKO, Carpinteria, CA, USA) using the warm bath method at 80 °C for 15 min. The primary antibody was rabbit anti-cathepsin K polyclonal antibody (1:2000, Abcum: ab19027) and incubated at 4 °C overnight. Secondary antibodies were EnVision+ System- HRP Labelled Polymer Anti-Rabbit (DAKO). For immunohistochemical reactions, Liquid DAB+ Substrate Chromogen System (DAKO) was used according to the manufacturer’s protocol. Mayer’s hematoxylin was used for nuclear staining.

The number and morphology of osteoclasts in the tissue sections were evaluated by hematoxylin and eosin staining and cathepsin K immunostaining (Figure 2). Cathepsin K-positive, multinucleated giant cells around the bone surface were identified as normal osteoclasts. On the other hand, cathepsin K-positive, giant multinucleated cells released from the bone surface, cathepsin K-positive mononuclear or small multinucleated cells on the bone surface, and no cathepsin-K positive multinucleated cells on the bone surface were regarded as suppressed osteoclasts.

## 3. Results

### 3.1. Impact of AR Drug Holidays on Treatment Outcomes

Table 1 shows the characteristics of 166 patients with MRONJ. Surgical therapy was performed in 139 patients and conservative therapy in 27. In the surgical therapy group, marginal resection (marginal mandibulectomy or partial maxillectomy) was performed in 133 of 139 patients, and only 6 patients underwent segmental mandibulectomy.

The cumulative cure rate was indicated by the presence or absence of drug holidays (Figure 3). There was no difference in treatment outcomes between the surgical and conservative treatment groups after a 3-month drug holiday.

The Cox regression analysis examined factors associated with treatment outcomes, and the univariate analysis demonstrated significantly poorer prognosis in cases with diabetes, low serum albumin, high serum creatinine, and conservative therapy (Table 2). Cases involving the mandible had poorer outcomes than those involving the maxilla, but the difference was not significant. Although there was no relationship between the stage and the healing rate, this may suggest that most of the patients underwent surgical therapy and that the operation was performed according to the stage. In a multivariate analysis in which these four factors and AR drug holiday were entered as covariates, three factors were significantly associated with poor treatment outcomes: mandibular bone development, low serum albumin, and conservative therapy (Table 3). Three-month AR drug holidays were not correlated with treatment outcomes in the univariate and multivariate analyses.

Since the treatment method was a major factor affecting the treatment outcomes, the same analysis was performed stratifying by the treatment method. In the 139 surgical cases, mandibular bone development and low serum albumin were associated with poor outcomes in the univariate analysis (Table 4). Multivariate analysis demonstrated that mandibular development and low serum albumin were associated with poor outcomes, although drug holidays for more than 3 months did not improve the healing rate (Table 5). In the 27 patients who underwent conservative surgery, no prognostic factors were found by univariate analysis, probably due to the small sample size (Table 6).

### 3.2. Histological Investigation for Recovery from Osteoclast Suppression and Drug Holiday

From the patients with osteoporosis with MRONJ receiving ARs, 139 patients underwent surgery. However, in many cases of marginal resection, viable bone samples could not be obtained because the necrotic bone was excised, and then the surrounding living bone was scraped away. As a result, a total of 43 specimens of living bone tissue were obtained and used for immunohistological examination. Table 7 summarizes the 43 patients who underwent histological investigation. Six patients were males and 37 were females, aged 59–92 (mean; 81.2) years. BP only was administrated in 30 patients, DMB only in 5 patients, and BP was changed to DMB in 8 patients. Surgery was performed without a drug holiday in 27 patients and 16 patients had drug holidays for 7 to 678 days. Only 2 patients did not have osteoclast suppression, whereas 17 patients had moderate suppression and 24 had severe suppression.

Figure 4 shows the relationship between drug holidays and osteoclast suppression. There were 31 patients with none or less than 3 months of drug holidays, of which 20 showed only suppressed osteoclasts, 6 showed both suppressed and normal osteoclasts, 4 showed neither suppressed nor normal osteoclasts, and only one showed normal osteoclasts without suppressed osteoclasts. Of the 8 patients with drug holidays for 3 to 12 months, 2 showed only suppressed osteoclasts, 4 showed both suppressed and normal osteoclasts, one showed neither suppressed nor normal osteoclasts, and one showed only normal osteoclasts. Of the 2 patients with drug holidays for 1 to 2 years, one had only suppressed osteoclasts, and one showed both suppressed and normal osteoclasts. Since the patients administered only DMB had no drug holidays or only short-term holidays, osteoclast suppression after long-term DMB withdrawal could not be investigated.

## 4. Discussion

In this study, we investigated the relationship between the AR drug holidays and treatment outcomes in osteoporosis patients receiving ARs who underwent surgical treatment, and histopathological examinations of osteoclast suppression were also conducted. The results indicated that osteoclast suppression was not recovered by 3-month drug holidays in patients administered BP or DMB, and 3-month drug holidays did not contribute to the improvement of the treatment outcomes.

According to Damm et al. [20], the skeletal system has a mixture of resting surfaces (osteocytes: 85%), resorbing surfaces (osteoclasts: 2%), and forming surfaces (osteoblasts: 10–12%). The lifespan of bone cells is several to 10 years, the lifespan of osteoclasts is 2 weeks, and the lifespan of osteoblasts is approximately 1 to 3 months. BP and DMB suppress bone resorption by different mechanisms, but both exert a bone resorption inhibitory effect by suppressing osteoclasts. Therefore, impaired bone remodeling is considered one of the causes of MRONJ onset. Approximately 50% of the BP taken into the serum is rapidly excreted from the kidney within a few hours, thus that the blood concentration decreases rapidly. However, BP specifically binds to the bone matrix, and the BP incorporated into the bone matrix is considered to remain in the bone until remodeling; its half-life ranges from several to 10 years. BP is released from the bone matrix in the acidic environment of the resorption fossa during the bone resorption process of osteoclasts, and the osteoclasts take up this to destroy the osteoclast skeleton, thereby inducing apoptosis and suppressing bone resorption. Osteoclasts under the influence of BP are observed as polynuclear, large osteoclasts detached from the bone surface [21].

In contrast, DMB taken up in serum circulates in the blood including blood vessels in the bone marrow. The half-life in blood is 25.4 days. Unlike the BP preparation, it does not show specific binding to the bone matrix and does not accumulate. Therefore, the action of DMB is considered to be more reversible than the BP preparation [22]. By specifically binding to RANKL, DMB inhibits the binding of RANKL to the receptor activator of nuclear factor kappa B (RANK) expressed on the surface of osteoclast precursor cells, suppresses differentiation into mature osteoclasts, and activates the existing mature osteoclasts. It also suppresses osteoclasts and survival. As a result, the total number of osteoclasts under the influence of DMB decreases, and small immature osteoclasts can be observed [23]. Since we wanted to investigate the difference between the period of drug holiday osteoclast inhibition in both drugs, we registered both BP and DMB cases as target cases this time. Although the number of DMB cases was small and a clear conclusion could not be drawn, we were able to suggest that osteoclasts do not recover even in DMB cases after several months of withdrawal.

Several conflicting opinions have been reported regarding the effects of AR drug holidays before tooth extraction. Otto et al. [24] demonstrated that zoledronic acid withdrawal before and after tooth extraction reduced the incidence of MRONJ in an animal experimental model. In contrast, Salgueiro et al. [25] stated that a 5-month drug holiday of the injectable BP drug does not affect the onset of MRONJ. Hasegawa et al. [13,26,27] also found in a multicenter study with a large number of cases that drug suspension did not affect the incidence of MRONJ after tooth extraction in osteoporosis patients receiving BP and high-dose ARs. In clinical practice, there is no report with a high level of evidence indicating the incidence of MRONJ decreased due to drug suspension before tooth extraction.

Regarding AR drug holidays during MRONJ treatment, the AAOMS and JSOMS position papers stated that it is desirable to withdraw ARs from the time of MRONJ diagnosis until the end of treatment [7,8]. Kim et al. [28] reported that in the MRONJ surgical treatment group, the treatment results of the 4-months or more drug holiday group were better than those of the non-drug group. Ramaglia et al. [29] also demonstrated that a drug holiday protocol promotes post-surgery healing, but the application must be determined for each patient’s condition. However, their papers are retrospective studies with a small number of cases and the level of evidence is low. In contrast, Hayashida et al. [30] collected a large number of MRONJ cases in a multicenter study and reported that AR drug holidays for approximately 3 months before surgery did not affect the treatment results of MRONJ surgery. Morishita et al. [18] reported that 2- to 6-month AR drug holidays before surgery did not affect treatment outcomes in osteoporosis patients with MRONJ. Otsuru et al. [31] also found that 2 to 4 months of preoperative high-dose AR drug holidays did not affect treatment outcomes in patients with cancer. As described above, the effectiveness of AR drug holidays during MRONJ treatment is also controversial, but there is no report with a high level of evidence that drug holidays improve MRONJ’s surgical results. In the present study, it was clarified that drug holidays of BP and DMB for several months did not recover the osteoclast suppression and did not affect the treatment outcome of MRONJ. Therefore, we believe that withdrawal of ARs should not be performed before tooth extraction or during MRONJ treatment.

This study has several limitations. Since the study was retrospective with a small number of cases, the criteria and duration of AR drug holidays differed depending on the case, and the patient background varied greatly, thus it is difficult to generalize the results obtained. The cases used in the histological examination were limited to those who underwent marginal resection or segmental resection, and selection bias may have occurred. For future research, we are considering increasing the number of cases, conducting more detailed examinations, and searching for local bone metabolism markers in molecular biology as well as a histological search.

## 5. Conclusions

This clinical study was conducted with patients using ARs for osteoporosis treatment and found that AR drug holidays for approximately 3 months neither recover osteoclast suppression nor affect treatment outcomes.

## Figures and Tables

**Figure 1 ijerph-19-10898-f001:**
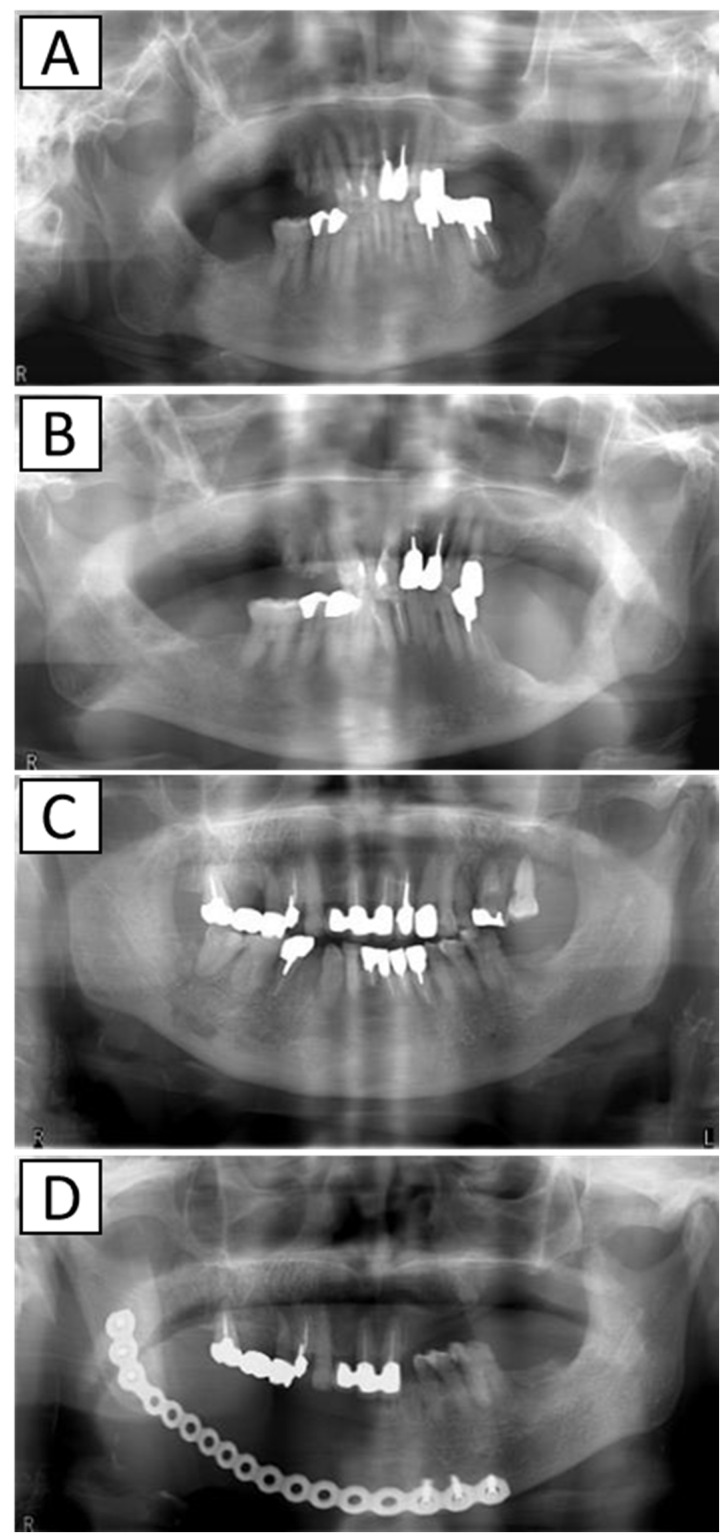
X-ray images before and after surgery. (**A**): Before marginal mandibulectomy, (**B**): after marginal mandibulectomy, (**C**): before segmental mandibulectomy, and (**D**): after segmental mandibulectomy.

**Figure 2 ijerph-19-10898-f002:**
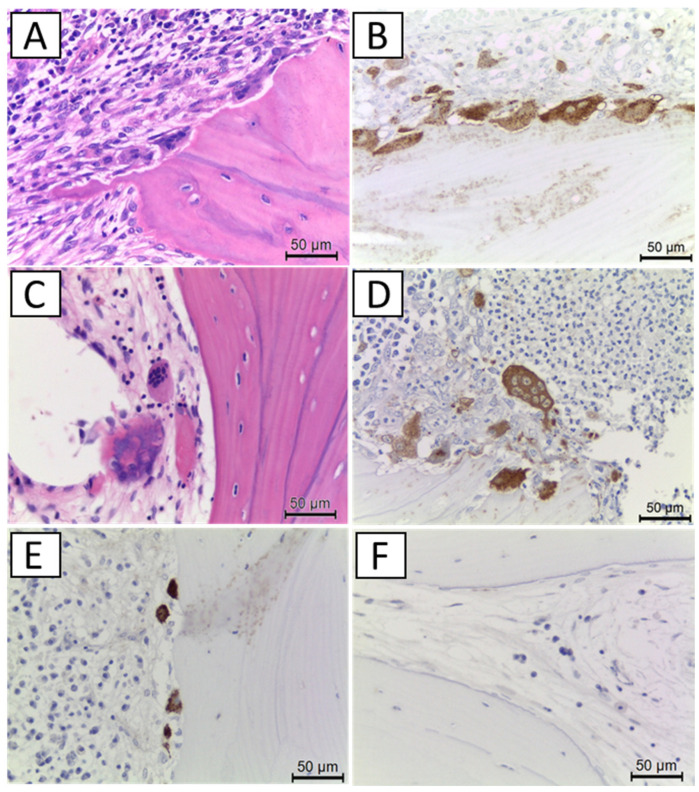
Histological findings of osteoclasts. (**A**): Normal osteoclasts (hematoxylin and eosin [HE] stain), (**B**): normal osteoclasts (cathepsin K stain), and (**C**): osteoclast suppression by BP. Giant multinuclear osteoclasts released from the bone surface (HE stain) and (**D**): osteoclast suppression by BP. Giant multinuclear osteoclasts released from the bone surface (cathepsin K stain) and (**E**): osteoclast suppression by DMB. Cathepsin-K positive, immature mononuclear or a small number of multinuclear cells and (**F**): osteoclast suppression by DMB. No cathepsin-K positive cells on the bone surface.

**Figure 3 ijerph-19-10898-f003:**
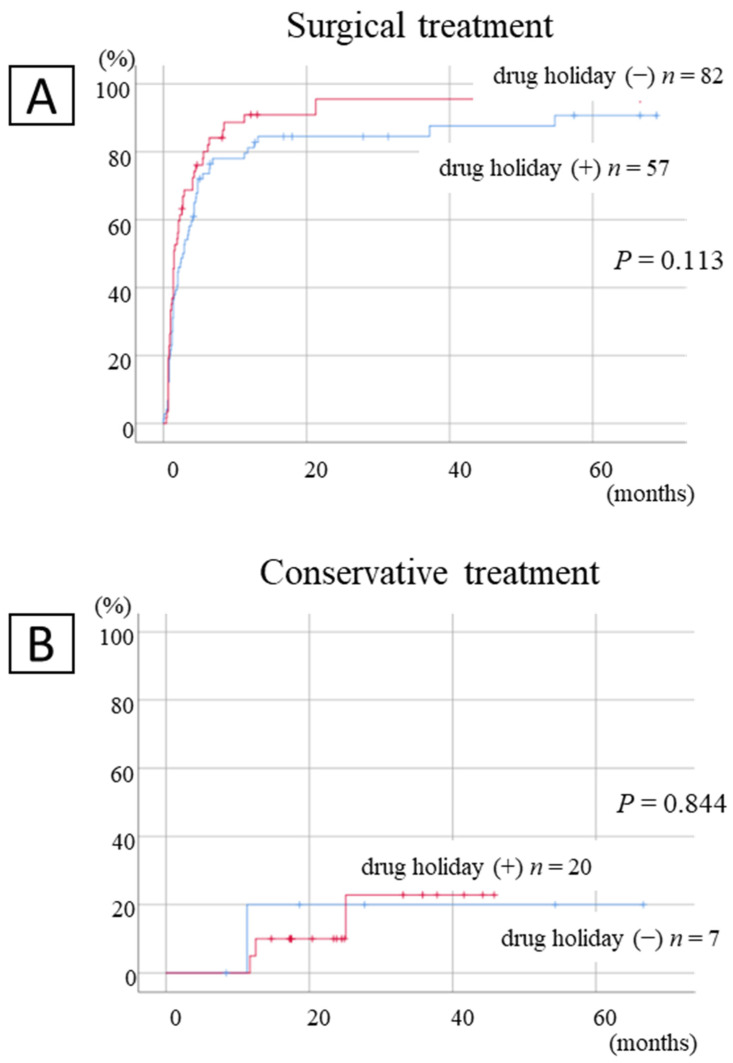
Cumulative healing rate by drug holidays. (**A**): Patients undergoing surgical treatments and (**B**): patients undergoing conservative treatments.

**Figure 4 ijerph-19-10898-f004:**
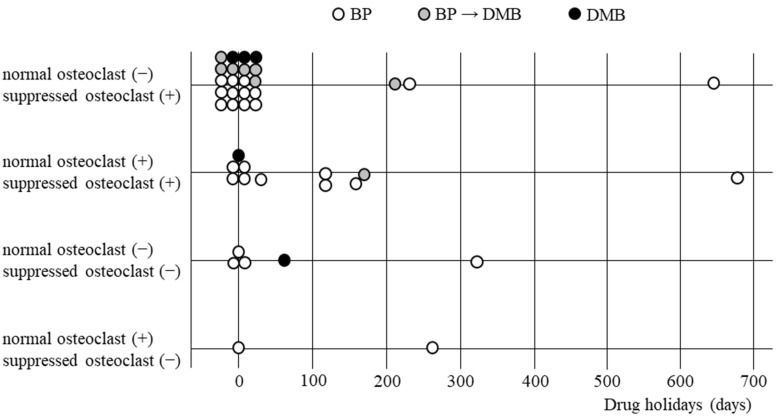
Relationship between drug holidays and osteoclast suppression. BP, bisphosphonate; DMB, denosumab.

**Table 1 ijerph-19-10898-t001:** Patient characteristics.

Variable		Conservative Therapy	Surgical Therapy
Sex	Male	4	17
	Female	23	122
Age		80.6 ± 9.20	78.5 ± 9.20
Site	Upper jaw	6	32
	Lower jaw	21	107
Stage	Stage 1	3	2
	Stage 2	16	99
	Stage 3	8	38
Sort of AR	BP	23	113
	Alendronate	19	72
	Risedronate	2	10
	Minodronate	1	8
	Etidronate	0	7
	Ibandronate	0	5
	Zoledronate	0	4
	Alendronate→minodronate	0	4
	Alendronate→risedronate	0	1
	Ibandronate→alencronate	0	2
	Unknown	1	0
	DMB	4	26
Duration of administration	<4 years	8	60
	≥4 years	16	70
	unknown	3	9
Corticosteroid	(−)	19	100
	(+)	8	9
Diabetes	(−)	19	123
	(+)	8	16
Serum albumin		3.62 ± 0.491	3.83 ± 0.491
Serum creatinine		0.635 ± 0.861	0.890 ± 0.861
Drug withdrawal for ≥3 months	(−)	7	82
	(+)	20	57
Surgical method	Marginal resection	-	133
	Segmental resection	-	6
Follow-up period (days)	Mean ± SD	447 ± 370	447 ± 370
Total		27	139

Abbreviation: AR: antiresorptive agent, BP: bisphosphonate, and DMB: denosumab.

**Table 2 ijerph-19-10898-t002:** Factors related to healing rate of MRONJ by univariate analysis (Total: 166 patients).

Variable		*p*-Value	HR	95% CI
Sex	Female/male	0.803	0.937	0.561–1.564
Age		0.255	0.990	0.973–1.007
Site of MRONJ	Lower jaw/upper jaw	0.058	0.679	0.456–1.013
Stage of MRONJ	Stage 3/2/1	0.946	0.988	0.703–1.389
Sort of AR	DMB/BP	0.461	1.167	0.774–1.760
Duration of administration	≧4 years/<4 years	0.338	0.838	0.583–1.204
Corticosteroid	(+)/(−)	0.480	0.867	0.583–1.288
Diabetes	(+)/(−)	0.028	0.535	0.306–0.935
Serum albumin		0.003	1.670	1.196–2.331
Serum creatinine		0.025	0.473	0.246–0.908
Drug withdrawal for ≥3 months	(+)/(−)	0.375	0.851	0.596–1.215
Treatment method	surgical/conservative	<0.001	14.126	5.177–38.543

Cox regression analysis: Abbreviation: AR: antiresorptive agent, BP: bisphosphonate, DMB: denosumab, HR: hazard ratio, and CI: confidence interval.

**Table 3 ijerph-19-10898-t003:** Factors related to healing rate of MRONJ by multivariate analysis (Total: 166 patients).

Variable		*p*-Value	HR	95% CI
Site of MRONJ	Lower jaw/upper jaw	0.045	0.650	0.427–0.990
Diabetes	(+)/(−)	0.371	0.757	0.411–1.393
Serum albumin		0.018	1.665	1.078–2.239
Serum creatinine		0.141	0.625	0.334–1.168
Drug withdrawal for ≥3 months	(+)/(−)	0.324	1.210	0.828–1.770
Treatment method	Surgical/conservative	<0.001	13.104	4.080–42.088

Cox proportional hazard model: Abbreviation: HR: hazard ratio and CI: confidence interval.

**Table 4 ijerph-19-10898-t004:** Factors related to healing rate of MRONJ by univariate analysis of 139 patients undergoing surgical therapy.

Variable		*p*-Value	HR	95% CI
Sex	Female/male	0.688	0.898	0.529–1.522
Age		0.551	0.995	0.977–1.013
Site	Lower jaw/upper jaw	0.019	0.614	0.409–0.922
Stage	Stage 3/2/1	0.600	0.901	0.609–1.332
Sort of AR	DMB/BP	0.198	1.355	0.853–2.151
Duration of administration	≧4 years/<4 years	0.543	0.892	0.617–1.289
Corticosteroid	(+)/(−)	0.337	0.819	0.545–1.231
Diabetes	(+)/(−)	0.317	0.745	0.418–1.326
Serum albumin		0.015	1.576	1.094–2.270
Serum creatinine		0.139	0.652	0.369–1.149
Drug withdrawal for ≥3 months	(+)/(−)	0.131	1.324	0.919–1.907
Surgical method	Segmental/marginal	0.502	1.327	0.581–3.030

Cox regression analysis: Abbreviation, AR: antiresorptive agent, BP: bisphosphonate, DMB: denosumab, HR: hazard ratio, and CI: confidence interval.

**Table 5 ijerph-19-10898-t005:** Factors related to healing rate of MRONJ by multivariate analysis of 139 patients undergoing surgical therapy.

Variable		*p*-Value	HR	95% CI
Site	Lower jaw/upper jaw	0.028	0.627	0.414–0.952
Serum albumin		0.029	1.502	1.042–2.166
Drug withdrawal for ≥3 months	(+)/(−)	0.190	1.287	0.882–1.877

Cox proportional hazard model: Abbreviation, HR: hazard ratio, and CI: confidence interval.

**Table 6 ijerph-19-10898-t006:** Factors related to healing rate of MRONJ by univariate analysis of 27 patients undergoing conservative therapy.

Variable		*p*-Value	HR	95% CI
Sex	Female/male	0.673	0.614	0.064–5.925
Age		0.915	1.006	0.897–1.129
Site	Lower jaw/upper jaw	0.981	1.028	0.106–9.924
Stage	Stage 3/2/1	0.597	0.646	0.128–3.259
Sort of AR	DMB/BP	0.273	2.386	0.504–11.284
Duration of administration	≧4 years/<4 years	0.394	40.942	0.008–207654
Corticosteroid	(+)/(−)	0.438	2.173	0.306–15.442
Diabetes	(+)/(−)	0.606	0.548	0.056–5.386
Serum albumin		0.650	1.473	0.277–7.830
Serum creatinine		0.254	0.047	0.000–9.067
Drug withdrawal for ≥3 months	(+)/(−)	0.943	0.920	0.095–8.932

Cox regression analysis: Abbreviation, AR: antiresorptive agent, BP: bisphosphonate, DMB: denosumab, HR: hazard ratio, and CI: confidence interval.

**Table 7 ijerph-19-10898-t007:** Summary of histologic findings.

Case	Sex	Age	Antiresorptive Agent	Administration Period	Duration of Drug Holiday	Histological Findings	
						Normal Osteoclasts	Suppressed Osteoclasts
1	Female	89	BP (alendronate)	3538	155	+	+
2	Female	59	BP (alendronate)	Unknown	0	+	+
3	Male	96	BP (minodronate)	More than 10 years	0	−	+
4	Female	81	BP (minodronate)	2435	640	−	+
5	Female	79	BP (alendronate)	644	224	−	+
6	Female	78	BP (alendronate)	About 2 years	0	−	+
7	Female	82	BP (alendronate)	2642	0	−	+
8	Male	89	BP (minodronate)	Unknown	0	−	+
9	Female	90	BP (ibandronate)	Unknown	Unknown	−	+
10	Female	92	BP (risedronate)	More than 5 years	0	−	+
11	Female	89	BP (alendronate)	4122	115	+	+
12	Female	89	BP (alendronate)	4122	115	+	+
13	Female	79	BP (alendronate)	Unknown	0	+	+
14	Female	77	BP (alendronate)	Unknown	0	−	−
15	Female	83	BP (alendronate)	2192	678	+	+
16	Female	87	BP (alendronate)	2748	261	+	−
17	Female	92	BP (ibandronate)	2657	0	−	+
18	Female	88	BP (risedronate)	Unknown	0	−	+
19	Female	87	BP (minodronate)	About 4 years	Unknown	+	+
20	Female	74	BP (alendronate)	2411	0	−	+
21	Female	84	BP (alendronate)	1292	0	−	+
22	Female	81	BP (alendronate)	161	0	+	+
23	Female	87	BP (alendronate)	Unknown	313	−	−
24	Male	68	BP (alendronate)	2031	0	−	−
25	Female	77	BP (alendronate)	1413	0	−	+
26	Male	80	BP (alendronate)	2944	0	+	+
27	Female	74	BP (alendronate)	922	0	+	−
28	Female	86	BP (alendronate)	1468	7	+	+
29	Female	82	BP (minodronate)	About 5 years	0	+	+
30	Female	82	BP (risedronate)	782	0	−	+
31	Female	87	BP (risedronate)→BP (alendronate)→DMB	1977	0	−	+
32	Male	79	BP (alendronate)→DMB	1996	0	−	+
33	Female	86	BP (alendronate)→DMB	1013	0	−	+
34	Female	91	BP (alendronate)→DMB	2588	0	−	+
35	Female	80	BP (risedronate)→BP (alendronate)→DMB	1371	206	−	+
36	Female	89	BP (alendronate)→DMB	3319	0	−	+
37	Female	70	BP (ibandronate)→DMB	731	36	−	+
38	Female	90	BP (zoledronate)→DMB	721	171	+	+
39	Male	65	DMB	361	68	−	−
40	Female	85	DMB	560	0	+	+
41	Female	65	DMB	Unknown	19	−	+
42	Female	59	DMB	184	61	−	+
43	Female	84	DMB	Unknown	13	−	+

Abbreviation: BP: bisphosphonate and DMB: denosumab.

## Data Availability

The datasets used and analyzed during the study are available from the corresponding author upon reasonable request.

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
