# Peer review of "Ineffectiveness of Antiresorptive Agent Drug Holidays in Osteoporosis Patients for Treatment of Medication-Related Osteonecrosis of the Jaw: Consideration from Immunohistological Observation of Osteoclast Suppression and Treatment Outcomes"

_ijerph, 2022, doi:10.3390/ijerph191710898_

Round 1
Reviewer 1 Report
This study described that whether antiresorptive agent drug holiday need or not on treatment outcome, in patients with MRONJ. It is including important information for drug holiday and osteoclast activity. However, the author should be more detailed explanation and adding the images.
1, In this article, the authors should delete Denosumab cases. Because, this pharmacological effect is different from BP.
2, The authors should described diagnostic criteria for MRONJ in Material and Method. Therefore, it should added more clinical symptom of MRONJ at first visit that is, pus discharge, bone exposure, alveolar abscess, paresthesia, ONJ induced maxillary sinusitis and more.
3, The definition of healing or non-healing is ambiguous. This ambiguous expression will be confused for readers. The authors should described classified criteria as healing or non-healing. Such as, when these wounds evaluated period. Therefore, the clinician should be re-evaluated wound condition at 3 months later. Because, only time will tell whether healing or non-healing is right.
4, In histopathological investigation, the authors should add osteoclast marker. At least one marker of osteoclast should show that is TRAP and/ or MMP-9 with immunohistochemistry test.
5, In osteoclast inhibition classification, it need images for Severe, Mild and None. And more explanation should add, respectively.
Author Response
To Reviewer #1
- In this article, the authors should delete Denosumab cases. Because, this pharmacological effect is different from BP.
(Reply)
Thank you for your comment. As mentioned in the Discussion, the mechanisms of BP and DMB are different. Since we wanted to investigate the difference between the period of drug holiday osteoclast inhibition in both drugs, we registered both BP and DMB cases as target cases this time. Although the number of DMB cases was small and a clear conclusion could not be drawn, we were able to suggest that osteoclasts do not recover even in DMB after several months of withdrawal.
- The authors should describe diagnostic criteria for MRONJ in Material and Method. Therefore, it should added more clinical symptom of MRONJ at first visit that is, pus discharge, bone exposure, alveolar abscess, paresthesia, ONJ induced maxillary sinusitis and more.
(Reply)
The symptoms at the first visit were not described, but according to the definition and staging of MRONJ, all cases had bone exposure, stage I had no infection symptoms, stage 2 had infection symptoms such as pus discharge, swelling, fistula, etc., and stage 3 showed symptoms such as extensive osteolysis and cutaneous fistulas.
Line 69-70: “Diagnostic criteria for MRONJ are based on the AAOMS Position Paper [7].” was added.
Line 70-71: “according to the criteria of AAOMS Position Paper” was inserted after “stage”.
- The definition of healing or non-healing is ambiguous. This ambiguous expression will be confused for readers. The authors should describe classified criteria as healing or non-healing. Such as, when these wounds evaluated period. Therefore, the clinician should be re-evaluated wound condition at 3 months later. Because, only time will tell whether healing or non-healing is right.
(Reply)
Line 96-97: The sentence “Patients whose symptoms disappeared once and were judged to be “healing" but later relapsed were classified as "non-healing".” was added, and the follow-up period was described in Table 1.
- In histopathological investigation, the authors should add osteoclast marker. At least one marker of osteoclast should show that is TRAP and/ or MMP-9 with immunohistochemistry test.
(Reply)
TRAP and cathepsin-K are usually used as osteoclast markers. Since TRAP may not be stained in decalcified specimens, all cases were investigated by cathepsin-K immunostaining in this study.
- In osteoclast inhibition classification, it needs images for Severe, Mild and None. And more explanation should add, respectively.
(Reply)
Figure 2 was revised.
Line 122-128: The description about osteoclast suppression was revise as follows.
Cathepsin K-positive, giant multinucleated cells released from the bone surface were regarded as osteoclast suppression by BP. Cathepsin K-positive mononuclear or small multinucleated cells on the bone surface were regarded as osteoclast suppression by DMB. Those in which only normal osteoclasts were observed were classified as no suppression, those in which only suppressed osteoclasts were observed or those with no cathepsin-K positive multinucleated cells on the bone surface were classified as severe suppression, and those in which both were observed were classified as moderate suppression.
Reviewer 2 Report
I sincerely congratulate you on an interesting study and a good report. Below are my comments:
- At first, you write that in this study "retrospectively histologically examined ... etc." (which I think is a kind of objective), and later that out of 166 patients, 43 had a histopathological examination done. You have to decide whether the study group comprises 166 or 43 patients. Then clearly define the objective or objectives of that study and make a paragraph from it (labeled as a separate section or last subsection of the Introduction).
- Please specify what you mean by "low-dose".
- State which specific antiresorptive agents were used. If they were various drugs, present (in Results) absolute numbers or percentages of patients from surgical and conservative groups treated with particular drugs. I only found a general breakdown for BP and Dmab. A heterogeneous group of BP should not be considered as one drug.
- In Fig. 1 group the before and after photos according to the patients, that is, according to the current markings, should be A, C, B, and D.
- In line 81 you write about the removal of healthy bone. What is your healthy bone margin? Always the same or was its size dependent on something?
- Why did you decide on two different p-values? By what test were they counted? (I only found it was a two-tailed one)
- Lines 97 to 99 are results, not methods.
- Explain the abbreviations from Table 1 in the table title or better in the table footer.
- In the text starting from 156 lines, you have given p-values ​​in parentheses. You also list them in Table 2. Such duplication of content is inappropriate. Please decide whether you want to present these values ​​in the text or in the table.
- In Table 3, the units of the administration period are not uniform - please correct it. Time without unit is days? This should be clarified.
- You admit that MRONJ is a recently discovered disease with dynamically changing treatment guidelines. I do not understand why (apart from the historical outline) you base the discussion on many articles older than 2-3 years.
- Make sure Conclusions accurately respond to the problem previously identified as objective (s).
- The opinions of different authors on the Conclusions section differ and I respect it if you disagree with me. If I were you, I would transfer the current content of this section to the discussion, and as conclusions, I would put 2-3 sentences of a concise answer to the research question.
- I am surprised that this is another work of this team, probably based partly on the same data. Why did the authors choose not to limit the number of articles based on a study registered under number 21021509?:
https://www.mdpi.com/1660-4601/19/12/7430/htm
https://www.mdpi.com/1660-4601/19/8/4624
https://www.nature.com/articles/s41598-022-15720-7
https://www.sciencedirect.com/science/article/pii/S1991790221001963
Author Response
To Reviewer #2
- At first, you write that in this study "retrospectively histologically examined ... etc." (which I think is a kind of objective), and later that out of 166 patients, 43 had a histopathological examination done. You have to decide whether the study group comprises 166 or 43 patients. Then clearly define the objective or objectives of that study and make a paragraph from it (labeled as a separate section or last subsection of the Introduction).
(Reply)
This research consists of two parts. One is to examine the relationship between the presence or absence of drug holiday and the treatment outcomes in 166 patients receiving low-dose ARs. The second is to confirm the withdrawal period of ARs and histological findings using 139 out of 166 cases who underwent surgery. However, in many cases of marginal resection, viable bone samples could not be obtained because the necrotic bone was excised, and then surrounding healthy bone was scraped away. As a result, a total of 43 specimens of living bone tissue were obtained.
Line 208-212: “From the patients with osteoporosis with MRONJ receiving low-dose ARs, 43 patients had marginal mandibulectomy, partial maxillectomy, or segmental mandibulectomy, with living bone observed in the resection margins were included in the study. Cases in which no tissue specimen of living bone could be obtained because the surrounding healthy bone was cut after the necrotic bone was removed and cases with resection margins of necrotic bone were excluded.” was moved and revised to “From the patients with osteoporosis with MRONJ receiving low-dose ARs, 139 patients underwent surgery. However, in many cases of marginal resection, viable bone samples could not be obtained because the necrotic bone was excised, and then the surrounding living bone was scraped away. As a result, a total of 43 specimens of living bone tissue were obtained and used for immunohistological examination.”.
- Please specify what you mean by "low-dose".
(Reply)
Line 67-69: The sentence “ARs used at doses for prevention of fracture in osteoporosis patients were defined as low-dose ARs in this study, regardless of the duration or total dose of administration.” was added.
- State which specific antiresorptive agents were used. If they were various drugs, present (in Results) absolute numbers or percentages of patients from surgical and conservative groups treated with particular drugs. I only found a general breakdown for BP and Dmab. A heterogeneous group of BP should not be considered as one drug.
(Reply)
Sort of antiresorptive agents was described in Tables 1 and 5.
- In Fig. 1 group the before and after photos according to the patients, that is, according to the current markings, should be A, C, B, and D.
(Reply)
Fig.1 and the legend were revised.
- In line 81 you write about the removal of healthy bone. What is your healthy bone margin? Always the same or was its size dependent on something?
(Reply)
Line 90-93: “We have not resected only necrotic bone but have included surrounding healthy bone.” was revised to “We have resected not only the sequestrum but some healthy bone around it. The extent of bone resection was planned based on preoperative CT images, but was ultimately determined based on the intraoperative findings such as the presence or absence of bleeding from the bone and changes in bone color.”.
- Why did you decide on two different p-values? By what test were they counted? (I only found it was a two-tailed one)
(Reply)
In Figure 2, the difference in treatment outcomes between the withdrawal and non-withdrawal groups was illustrated by the Kaplan-Meier method and analyzed by the log-rank test. On the other hand, in Tables 2-4, since continuous variables are included as independent variables, Cox regression analysis was used for the difference in treatment outcomes between the two groups.
The test method was added in Tables 2-4.
- Lines 97 to 99 are results, not methods.
(Reply)
Line 109: “From the patients with osteoporosis with MRONJ receiving low-dose ARs, 43 patients had marginal mandibulectomy, partial maxillectomy, or segmental mandibulectomy, with living bone observed in the resection margins were included in the study. Cases in which no tissue specimen of living bone could be obtained because the surrounding healthy bone was cut after the necrotic bone was removed and cases with resection margins of necrotic bone were excluded.” was deleted.
Line 205-209: “From the patients with osteoporosis with MRONJ receiving low-dose ARs, 139 patients underwent surgery. However, in many cases of marginal resection, viable bone samples could not be obtained because the necrotic bone was excised, and then the surrounding living bone was scraped away. As a result, a total of 43 specimens of living bone tissue were obtained and used for immunohistological examination.” was added.
- Explain the abbreviations from Table 1 in the table title or better in the table footer.
(Reply)
The abbreviations were added in Table 1 footer.
- In the text starting from 156 lines, you have given p-values in parentheses. You also list them in Table 2. Such duplication of content is inappropriate. Please decide whether you want to present these values in the text or in the table.
(Reply)
The p-values in the text were deleted.
- In Table 3, the units of the administration period are not uniform - please correct it. Time without unit is days? This should be clarified.
(Reply)
The unit (days) of the administration period and duration of the drug holiday were added in Table 5.
- You admit that MRONJ is a recently discovered disease with dynamically changing treatment guidelines. I do not understand why (apart from the historical outline) you base the discussion on many articles older than 2-3 years.
(Reply)
Lines 232-251 Lines 277-289, and References #17-20 and #25-26 from the previous version were deleted.
- Make sure Conclusions accurately respond to the problem previously identified as objective (s).
(Reply)
Line 306-308: The description of the Conclusion was revised for brevity.
- The opinions of different authors on the Conclusions section differ and I respect it if you disagree with me. If I were you, I would transfer the current content of this section to the discussion, and as conclusions, I would put 2-3 sentences of a concise answer to the research question.
(Reply)
Line 306-308: The description of the Conclusion was revised for brevity.
- I am surprised that this is another work of this team, probably based partly on the same data. Why did the authors choose not to limit the number of articles based on a study registered under number 21021509?:
https://www.mdpi.com/1660-4601/19/12/7430/htm
https://www.mdpi.com/1660-4601/19/8/4624
https://www.nature.com/articles/s41598-022-15720-7
https://www.sciencedirect.com/science/article/pii/S1991790221001963
(Reply)
https://www.mdpi.com/1660-4601/19/12/7430/htm
This paper describes maxillary sinusitis associated with MRONJ.
https://www.mdpi.com/1660-4601/19/8/4624
This paper describes the relationship among drug holidays, sequestrum separation, and treatment outcomes in MRONJ patients with malignant tumor receiving high-dose AR therapy. This is an image-based study without a histopathological review.
https://www.nature.com/articles/s41598-022-15720-7
This paper described the relationship among drug holidays, sequestrum separation, and treatment outcomes in MRONJ patients with osteoporosis receiving low-dose AR therapy. This is an image-based study without a histopathological review.
https://www.sciencedirect.com/science/article/pii/S1991790221001963
This paper describes segmental mandibulectomy for MRONJ.
Although some patients overlap in some of these papers, the purposes and subject cases of this study are different, and these papers are completely different from the current study.
Reviewer 3 Report
Dear authors, thank you for the study. I have two suggestions: please change the title "Low-dose Antiresorptive Agent Drug Holidays and Recovery from Osteoclast Suppression: Histopathological Investigation and Effect on Treatment Outcomes". The title is not agreed upon at the beginning, nor does it make much sense to emphasize the routine and classical method of light microscopy used for histological analysis.
Also, in the description of microphotographs, it is necessary to indicate 1) magnification and expand the description of the findings. On the microphotographs themselves, arrows for the selected described elements, and bar scales.
Author Response
To Reviewer #3
- I have two suggestions: please change the title "Low-dose Antiresorptive Agent Drug Holidays and Recovery from Osteoclast Suppression: Histopathological Investigation and Effect on Treatment Outcomes". The title is not agreed upon at the beginning, nor does it make much sense to emphasize the routine and classical method of light microscopy used for histological analysis.
(Reply)
The title was revised to “Ineffectiveness of Low-dose Antiresorptive Agent Drug Holidays for Treatment of Medication-Related Osteonecrosis of the Jaw: Consideration from Immunohistological Observation of Osteoclast Inhibition and Treatment Outcomes”.
- Also, in the description of microphotographs, it is necessary to indicate 1) magnification and expand the description of the findings. On the microphotographs themselves, arrows for the selected described elements, and bar scales.
(Reply)
Figure 2 was revised, and magnification and description of the findings were added.
Reviewer 4 Report
There is no consensus regarding the withdrawal period of ARs for the prevention of MRONJ. This study focuses on osteoclasts and analyzes the relationship with the withdrawal period of ARs, which is considered to be clinically important. However, I think there are some points that should be taken into consideration before publishing this manuscript.
Line 6: Masahiro Umeda?
Line 26: antiresorptive agents?
Lines 64-65: How did the authors choose surgical therapy or conservative therapy? Surgical cases may have severe MRONJ and conservative cases may have mild MRONJ. Did the authors choose based on guidelines of MRONJ? Also, is there no selection bias?
Lines 96-121: The authors focused on osteoclasts. However, I think it is also necessary to analyze osteoblasts or osteoprotegerin. Please consider about this point.
Table 1: Conservative therapy?
Table 1: Total number of sort of AR in the conservative therapy group are 26. Is it correct?
Line 178: Table 4?
Line 179: The description of Table 4 is not appropriate. Please check.
Line 182: Table 5 instead of Table 3?
Line 182: There are 43 cases of histologic findings. However, I can't understand how to choose 43 cases in surgical treatment cases (N=139). Please explain about this point.
Table 4: female?
Table 5: female?
4. Discussion: I think the content of the discussion is redundant and not a proper consideration of the results. For example, lines 232-251 are content of Introduction section and not necessary for Discussion section. Also, lines 277-289 are not necessary for Discussion section. Please revise the Discussion section.
Author Response
To Reviewer #4
- Line 6: Masahiro Umeda?
(Reply)
Line 7: “Masahito Umeda” was corrected to “Masahiro Umeda”.
- Line 26: antiresorptive agents?
(Reply)
Line 27: “ntiresorptive agents” was corrected to “antiresorptive agents”.
- Lines 64-65: How did the authors choose surgical therapy or conservative therapy? Surgical cases may have severe MRONJ and conservative cases may have mild MRONJ. Did the authors choose based on guidelines of MRONJ? Also, is there no selection bias?
(Reply)
Line 74-76 “We have made surgical therapy the first choice for MRONJ regardless of the MRONJ stage, but conservative therapy was selected when surgery was not possible due to the patient's wishes or poor general condition.” was added.
- Lines 96-121: The authors focused on osteoclasts. However, I think it is also necessary to analyze osteoblasts or osteoprotegerin. Please consider about this point.
(Reply)
As you pointed out, in order to observe bone metabolism, it is necessary to observe not only osteoclasts but also osteoblasts. However, since BP and DMB are drugs that act on osteoclasts, this study focused on osteoclasts.
- Table 1: Conservative therapy?
(Reply)
“Consurvative therapy” was corrected to “Conservative therapy” in Table 1.
- Table 1: Total number of sort of AR in the conservative therapy group are 26. Is it correct?
(Reply)
Table 1 was corrected.
- Line 178: Table 4?
(Reply)
Table 3 was a mistake in Table 5, Table 4 was a mistake in Table 3, and Table 5 was a mistake in Table 4.
- Line 179: The description of Table 4 is not appropriate. Please check.
(Reply)
Table 3 was a mistake in Table 5, Table 4 was a mistake in Table 3, and Table 5 was a mistake in Table 4.
- Line 182: Table 5 instead of Table 3?
(Reply)
Table 3 was a mistake in Table 5, Table 4 was a mistake in Table 3, and Table 5 was a mistake in Table 4.
- Line 182: There are 43 cases of histologic findings. However, I can't understand how to choose 43 cases in surgical treatment cases (N=139). Please explain about this point.
(Reply)
This research consists of two parts. One is to examine the relationship between the presence or absence of drug holiday and the treatment outcomes in 166 patients receiving low-dose ARs. The second is to confirm the withdrawal period of ARs and histological findings using 139 out of 166 cases who underwent surgery. However, in many cases of marginal resection, viable bone samples could not be obtained because the necrotic bone was excised, and then surrounding healthy bone was scraped away. As a result, a total of 43 specimens of living bone tissue were obtained.
Line 109: “From the patients with osteoporosis with MRONJ receiving low-dose ARs, 43 patients had marginal mandibulectomy, partial maxillectomy, or segmental mandibulectomy, with living bone observed in the resection margins were included in the study. Cases in which no tissue specimen of living bone could be obtained because the surrounding healthy bone was cut after the necrotic bone was removed and cases with resection margins of necrotic bone were excluded.” was deleted.
Line 208-212: “From the patients with osteoporosis with MRONJ receiving low-dose ARs, 139 patients underwent surgery. However, in many cases of marginal resection, viable bone samples could not be obtained because the necrotic bone was excised, and then the surrounding living bone was scraped away. As a result, a total of 43 specimens of living bone tissue were obtained and used for immunohistological examination.” was added.
- Table 4: female?
(Reply)
“Femela” was corrected to “Female” in Table 4.
- Table 5: female?
(Reply)
“Femela” was corrected to “Female” in Table 5.
- Discussion: I think the content of the discussion is redundant and not a proper consideration of the results. For example, lines 232-251 are content of Introduction section and not necessary for Discussion section. Also, lines 277-289 are not necessary for Discussion section. Please revise the Discussion section.
(Reply)
Lines 232-251 Lines 277-289, and References #17-20 and 25-26 from the previous version were deleted.
Round 2
Reviewer 1 Report
I edited reply with Red Bold in word file.
I look forward to your response.

Author Response
- In this article, the authors should delete Denosumab cases. Because, this pharmacological effect is different from BP.
(Reply)
Thank you for your comment. As mentioned in the Discussion, the mechanisms of BP and DMB are different. Since we wanted to investigate the difference between the period of drug holiday osteoclast inhibition in both drugs, we registered both BP and DMB cases as target cases this time. Although the number of DMB cases was small and a clear conclusion could not be drawn, we were able to suggest that osteoclasts do not recover even in DMB after several months of withdrawal.
OK, could you describe above in discussion?
(Reply)
Line 329-334: The following sentences were added.
Since we wanted to investigate the difference between the period of drug holiday osteoclast inhibition in both drugs, we registered both BP and DMB cases as target cases this time. Although the number of DMB cases was small and a clear conclusion could not be drawn, we were able to suggest that osteoclasts do not recover even in DMB cases after several months of withdrawal.
- The authors should describe diagnostic criteria for MRONJ in Material and Method. Therefore, it should add more clinical symptoms of MRONJ at the first visit that is, pus discharge, bone exposure, alveolar abscess, paresthesia, ONJ induced maxillary sinusitis, and more.
(Reply)
The symptoms at the first visit were not described, but according to the definition and staging of MRONJ, all cases had bone exposure, stage I had no infection symptoms, stage 2 had infection symptoms such as pus discharge, swelling, fistula, etc., and stage 3 showed symptoms such as extensive osteolysis and cutaneous fistulas.
Line 71-72: “Diagnostic criteria for MRONJ are based on the AAOMS Position Paper [7].” was added.
Line 73: “according to the criteria of AAOMS Position Paper” was inserted after “stage”.
- Thank you for your editing. Could you edit relation between outcome and stage definition. If no relation that, AAOMS stage definition will be meaningless. Please give us your results and discussion.
(Reply)
Line 174-176: The following sentence was added.
Although there was no relationship between the stage and the healing rate, this may suggest that most of the patients underwent surgical therapy and that the operation was performed according to the stage.
- It is not sufficient to define cathepsin K expression with immunostaining as osteoclast activity. The authors should add quantitative way.
(Reply)
Since no quantitative observations were made this time, the classification of osteoclast activity (severe-, moderate-, or no suppression) was deleted.
The descriptions in “Materials and methods”, Table 5, and Figure 4 were revised.
Line 125-128: “Cathepsin K-positive, giant multinucleated cells released from the bone surface were regarded as osteoclast suppression by BP. Cathepsin K-positive mononuclear or small multinucleated cells on the bone surface were regarded as osteoclast suppression by DMB. Those in which only normal osteoclasts were observed were classified as no suppression, those in which only suppressed osteoclasts were observed or those with no cathepsin-K positive multinucleated cells on the bone surface were classified as severe suppression, and those in which both were observed were classified as moderate suppression.” was revised to “On the other hand, cathepsin K-positive, giant multinucleated cells released from the bone surface, cathepsin K-positive mononuclear or small multinucleated cells on the bone surface, and no cathepsin-K positive multinucleated cells on the bone surface were regarded as suppressed osteoclast.”.
Line 259-266:
The sentences “There were 31 patients with none or less than 3 months of drug holidays, of which 20 had severe osteoclast suppression, 10 had moderate suppression, and only 1 was judged to have no suppression. Of the 8 patients with drug holidays for 3 to 12 months, 2 had severe osteoclast suppression, 5 had moderate suppression, and 1 patient had no suppression. Of the 2 patients with drug holidays for 1 to 2 years, 1 had severe osteoclast suppression and 1 had moderate suppression.” were revised to “There were 31 patients with none or less than 3 months of drug holidays, of which 20 showed only suppressed osteoclasts, 6 showed both suppressed and normal osteoclasts, 4 showed neither suppressed nor normal osteoclasts, and only one showed normal osteoclasts without suppressed osteoclasts. Of the 8 patients with drug holidays for 3 to 12 months, 2 showed only suppressed osteoclasts, 4 showed both suppressed and normal osteoclasts, one showed neither suppressed nor normal osteoclasts, and one showed only normal osteoclasts. Of the 2 patients with drug holidays for 1 to 2 years, one had only suppressed osteoclasts, and one showed both suppressed and normal osteoclasts. “.
Table 7 and Figure 4 were revised.
Reviewer 2 Report
I would like to thank the editors and authors for the opportunity to re-evaluate the manuscript.
1. My first remark was not taken into account. There is still no Objective or Objectives paragraph before the methodology (neither as a separate section, nor as an introduction subsection).
2. The authors try to avoid answering my second comment. They use the term "low-dose" in the title of the manuscript, which later appears consistently throughout the body of the text. I understand this is about prophylactic doses. That is? The reader still cannot know what the authors mean. I urge the authors to explain in detail the dosing of ARs in their study in the body of the manuscript.
3. I thank the authors for specifying the types of drugs.
4. Thanks to the authors for improving Fig. 1.
5. I would like to thank the authors for explaining the adopted rules for determining the scope of the resection margin.
6. The authors still do not explain in the text of the manuscript why they use two different p-values. Typically, this practice is intended to mislead the reader. So that the reader cannot accuse the authors of this, it is necessary to explain where the different p-values ​​come from.
7. Thanks to the authors for the correction.
8. I thank the authors for adding explanations of the abbreviations.
9. Thanks to the authors for removing duplicate information.
10. Thanks to the authors for the corrections on the timeframe.
11. I thank the authors for removing old references.
12. and 13. I thank the authors for correcting the Conclusions.
14. Thanks to the authors for the explanation. In order for readers to have no doubts in this matter, in the Methods section, I suggest adding information about which articles are partly based on the same research group, very briefly adding what they refer to and providing references to them.
Summing up, the Authors have improved most of the issues I have identified. I would appreciate the opportunity to re-evaluate the manuscript after the next revision.
Author Response
- My first remark was not taken into account. There is still no Objective or Objectives paragraph before the methodology (neither as a separate section, nor as an introduction subsection).
(Reply)
Line 57-60:
“This study retrospectively histologically examined whether withdrawal recovered osteoclast suppression and whether there were clinical benefits for withdrawal in patients with MRONJ receiving low-dose BP and DMB.” was revised to “The purpose of this study is to examine whether withdrawal of ARs recovers osteoclast suppression and whether there are clinical benefits for withdrawal in osteoporosis patients with MRONJ receiving BP and DMB.”.
- The authors try to avoid answering my second comment. They use the term "low-dose" in the title of the manuscript, which later appears consistently throughout the body of the text. I understand this is about prophylactic doses. That is? The reader still cannot know what the authors mean. I urge the authors to explain in detail the dosing of ARs in their study in the body of the manuscript.
(Reply)
ARs used at doses for prevention of fracture in osteoporosis patients were defined as low-dose ARs in this study, regardless of the duration or total dose of administration. However, this description may be confusing for the reader. So we removed the term “low-dose”.
Line 2-3: “Ineffectiveness of Low-dose Antiresorptive Agent Drug Holidays for Treatment of -----” was revised to “Ineffectiveness of Antiresorptive Agent Drug Holidays in Osteoporosis Patients for Treatment of ----“.
Line 13: “low-dose” was deleted.
Line 18: “low-dose” was deleted.
Line 38: “low-dose” was deleted.
Line 44: “low-dose” was deleted.
Line 60-61: “in patients with MRONJ receiving low-dose BP and DMB” was revised to “in osteoporosis patients with MRONJ receiving BP and DMB”.
Line 65: “due to low-dose ARs” was deleted.
Line 67: “ARs used at doses for prevention of fracture in osteoporosis patients were defined as low-dose ARs in this study, regardless of the duration or total dose of administration.” was deleted.
Line 247: “low-dose” was deleted.
Line 301: “in patients receiving low-dose ARs” was revised to “in osteoporosis patients receiving ARs”.
Line 343: “in patients receiving low-dose BP” was revised to “in osteoporosis patients receiving BP”.
Line 357: “low-dose” was deleted.
Line 358: “osteoporosis“ was inserted before “patients”.
Line 373: “low-dose” was deleted.
- The authors still do not explain in the text of the manuscript why they use two different p-values. Typically, this practice is intended to mislead the reader. So that the reader cannot accuse the authors of this, it is necessary to explain where the different p-values come from.
(Reply)
The p-values in Tables 2 and 3 describe both univariate and multivariate analyses. To avoid confusing the reader, we have divided the tables into univariate and multivariate analyses.
- Thanks to the authors for the explanation. In order for readers to have no doubts in this matter, in the Methods section, I suggest adding information about which articles are partly based on the same research group, very briefly adding what they refer to and providing references to them.
(Reply)
The next sentences and references were added.
Line 67-71: Some of the cases used in this study overlap with some of the cases used in our group's previous reports on treatment methods for MRONJ, such as reports on treatment methods for maxillary MRONJ, reports on the relationship between drug holiday and sequestrum separation, and reports on segmental mandibulectomy.
Reviewer 4 Report
The authors have revised the manuscripts according to our reviewers' comments. However, I think there are some points that should be taken into consideration before publishing this manuscript.
Lines 273-278: The results of this study indicated that osteoclast suppression was not recovered by 3-month drug holidays in patients administered BP or DMB, and 3-month drug holidays did not contribute to the improvement of the treatment outcomes. Based on these results, I think it would be better to describe how the authors think about suspension of BP or DMB in Discussion section.
Author Response
To Reviewer #4
Lines 273-278: The results of this study indicated that osteoclast suppression was not recovered by 3-month drug holidays in patients administered BP or DMB, and 3-month drug holidays did not contribute to the improvement of the treatment outcomes. Based on these results, I think it would be better to describe how the authors think about suspension of BP or DMB in Discussion section.
(Reply)
Line 360-364: The next sentences were added in the Discussion section.
In the present study, it was clarified that drug holidays of BP and DMB for several months did not recover the osteoclast suppression and did not affect the treatment outcome of MRONJ. Therefore, we believe that withdrawal of ARs should not be performed before tooth extraction or during MRONJ treatment.